# Nuclear membrane protein Lem2 regulates nuclear size through membrane flow

Kazunori Kume [1,2,5], Helena Cantwell [2,5], Alana Burrell [3] & Paul Nurse[2,4]

The size of the membrane-bound nucleus scales with cell size in a wide range of cell types but the mechanisms determining overall nuclear size remain largely unknown. Here we investigate the role of fission yeast inner nuclear membrane proteins in determining nuclear size, and propose that the Lap2-Emerin-Man1 domain protein Lem2 acts as a barrier to membrane flow between the nucleus and other parts of the cellular membrane system. Lem2 deletion increases membrane flow into and out of the nuclear envelope in response to changes in membrane synthesis and nucleocytoplasmic transport, altering nuclear size. The endoplasmic reticulum protein Lnp1 acts as a secondary barrier to membrane flow, functionally compensating for lack of Lem2. We propose that this is part of the mechanism that maintains nuclear size proportional to cellular membrane content and thus to cell size. Similar regulatory principles may apply to other organelles in the eukaryotic subcellular membrane network.

[1] Hiroshima Research Center for Healthy Aging, Department of Molecular Biotechnology, Graduate School of Advanced Sciences of Matter, Hiroshima University, Higashi-Hiroshima 739-8530, Japan. [2] Cell Cycle Laboratory, The Francis Crick Institute, London NW1 1AT, UK. [3] Electron Microscopy Science Technology Platform, The Francis Crick Institute, London NW1 1AT, UK. [4] Laboratory of Yeast Genetics and Cell Biology, Rockefeller University, New York, NY 10065, USA. [5] These authors contributed equally: Kazunori Kume, Helena Cantwell. Correspondence and requests for materials should be addressed to K.K. (email: kume513@hiroshima-u.ac.jp)

The size of the nucleus scales with cell size in cell types ranging from yeast to animal cells[1–3]. The nucleus exists as part of a subcellular membrane network, with the nuclear envelope being continuous with the endoplasmic reticulum (ER)[4,5]. Despite much study of the molecular mechanisms of membrane growth in eukaryotic cells[6], the controls that regulate the overall size of the nucleus remain largely unknown.

In the fission yeast Schizosaccharomyces pombe, the nuclear volume to cell volume ratio (N/C ratio) is maintained constant as cells increase in size both throughout the cell cycle and across a 35-fold range of cell volumes[1]. Similar N/C ratios are observed in widely divergent yeast species[3] suggesting that maintenance of a particular ratio is important for cell physiology. Studies of multinucleate cells indicate that the amount of cytoplasm surrounding a nucleus determines its size[1,7,8]. In vitro experiments with Xenopus egg extracts[9,10] and a genetic screen in fission yeast[11] have implicated nuclear lamina components, nucleocytoplasmic transport, and overall lipid biosynthesis in nuclear size control. Nuclear lamin proteins which are lacking in yeasts have been implicated in nuclear size control in metazoans[9,10] and

**Fig. 1** Lem2 restricts nuclear size enlargement independently of its chromatin-binding activity. **a** N/C ratio of wild type (WT), rae1-167, rae1-167 lem2Δ, and rae1-167 lem2^hehΔ cells grown at 25 °C then shifted to the indicated temperature for 4 h (n > 30 cells). Two-tailed t-tests: ****P < 0.0001, rae1-167 (n = 64 cells) against rae1-167 lem2Δ (n = 35 cells), t = 4.299, degrees of freedom = 97; rae1-167 lem2^hehΔ (36 °C) (n = 31 cells) against rae1-167 (36 °C) (n = 64 cells), P = 0.0863, t = 1.741, degrees of freedom = 93. In lem2^hehΔ cells, the N-terminal helix-extension-helix chromatin-binding region of Lem2 is deleted. In box-and-whiskers diagrams, boxes indicate median and upper and lower quartile and whiskers indicate range of data. The corresponding dot plot is available in Supplementary Fig. 9a. **b** Images of the nuclear envelope (Cut11-GFP, green) of wild type (WT), rae1-167 and rae1-167 lem2Δ cells grown at 25 °C then shifted to the indicated temperature for 4 h. Maximum intensity projections shown. Scale bar: 5 μm. **c** Images of the nuclear envelope (Cut11-GFP, green) and chromatin (Hht1-mRFP, magenta) of rae1-167 lem2Δ cells grown at 25 °C then shifted to the indicated temperature for 2 h. Maximum intensity projections shown. Scale bar: 5 μm

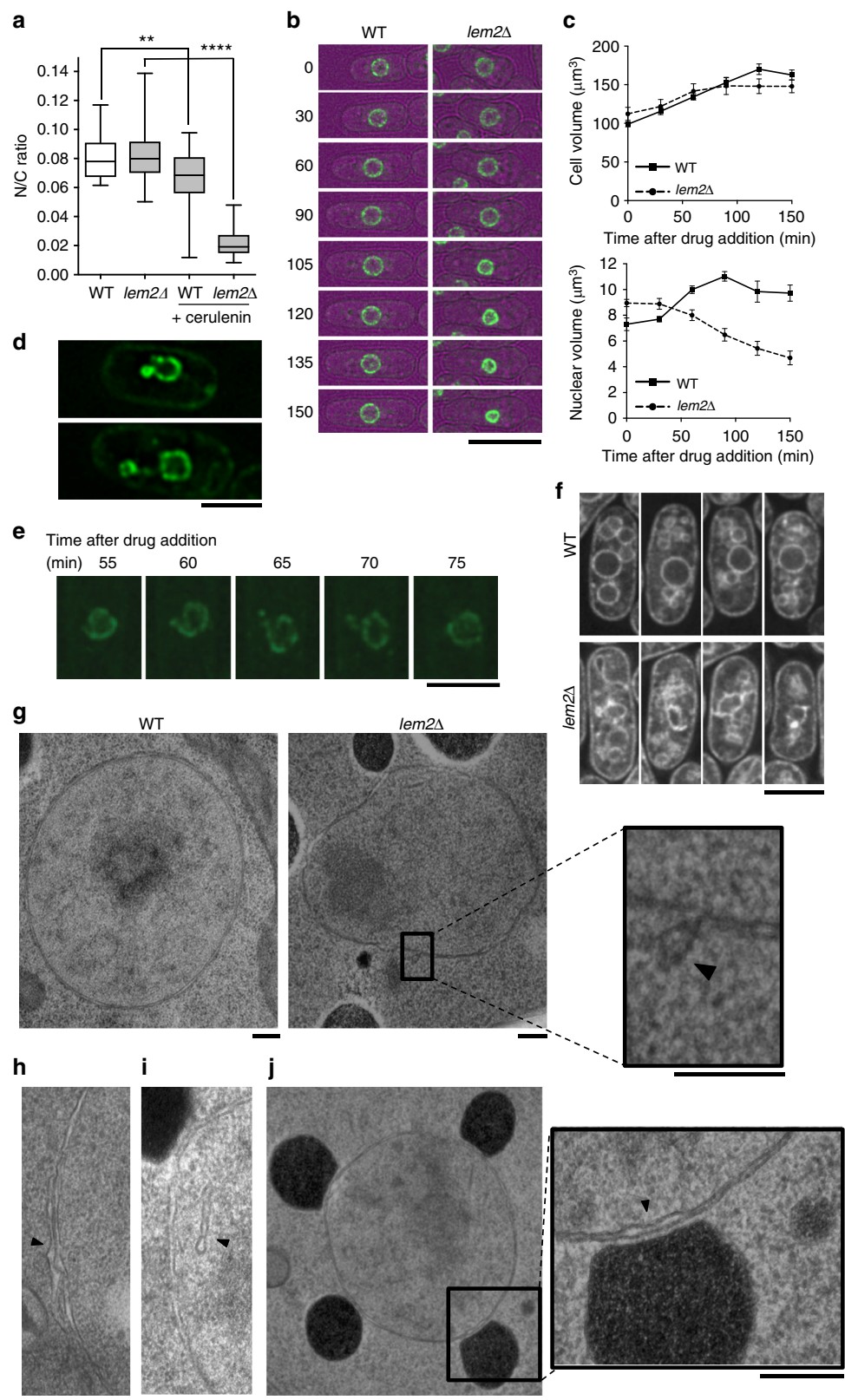

underlie the nuclear envelope, but the roles of other proteins associated with the nuclear membrane in this process have not been examined.

Here, we assess the contribution of inner nuclear membrane proteins to the maintenance of the N/C ratio in fission yeast. We demonstrate that deletion of Lap2-Emerin-Man1 (LEM) domain protein Lem2, but not that of other inner nuclear membrane proteins, augments nuclear size enlargement phenotypes resulting from perturbation of nucleocytoplasmic transport. We show that Lem2 deletion leads to nuclear shrinkage, accompanied by nuclear envelope blebbing, following perturbation of membrane synthesis. We propose that Lem2 forms part of a nuclear size

**Fig. 2** Cerulenin treatment causes shrinkage of *lem2Δ* interphase nuclei. **a** N/C ratio of wild type (WT) and *lem2Δ* cells grown with or without 10 µg/ml cerulenin for 3 h (25 °C) ($n > 30$ cells). Two-tailed Mann–Whitney test: ****$P < 0.0001$, **$P < 0.01$, WT ($n = 35$ cells), WT+cerulenin ($n = 44$ cells), *lem2Δ* +cerulenin ($n = 34$ cells). In box-and-whiskers diagrams, boxes indicate median, and upper and lower quartile, and whiskers indicate range of data. The corresponding dot dot plot is available in Supplementary Fig. 9b. **b** Stills from timelapse microscopy of a WT and a *lem2Δ* cell, treated with 10 µg/ml cerulenin for indicated time (min) (32 °C). Cell (brightfield, magenta), nuclear envelope (Cut11-GFP, green) maximum intensity projection. Scale bar: 10 µm. Supplementary Movies 1–4 display full timecourse and further cells. **c** Cell and nuclear volume of WT (solid line) and *lem2Δ* (dashed line) cells from experiment described in **b** ($n = 15$ interphase cells per strain). Values are shown as mean ± s.e.m. **d** Images of *lem2Δ* cells treated with 10 µg/ml cerulenin for 120 min (25 °C). Nuclear envelope (Cut11-GFP, green) single z-slice. Scale bar: 5 µm. **e** Nucleus of a *lem2Δ* cell treated with 10 µg/ml cerulenin (32 °C), at 5 min time intervals after drug addition that shows blebbing. Nuclear envelope (Cut11-GFP, green) maximum intensity projection. Scale bar: 5 µm. Movies displaying full timecourse and further cells in Supplementary Movies 5–8. **f** Images of WT and *lem2Δ* cells, treated with 10 µg/ml cerulenin for 90 min (32 °C). Cellular membranes (DiOC$_6$, grayscale) single z-slice. Scale bar: 5 µm. **g** Transmission electron microscopy image of a representative WT and a representative *lem2Δ* nucleus treated with 10 µg/ml cerulenin for 90 min. Irregular nuclear shape, irregular distance between inner and outer nuclear membranes, and apparent nuclear envelope blebbing event (black arrowhead) are observed in *lem2Δ* and not in WT. Scale bars: 250 nm. **h–j** Transmission electron microscopy images of *lem2Δ* cells treated with 10 µg/ml cerulenin for 90 min at 32 °C. Apparent nuclear envelope blebbing event (**h**), intranuclear membranous structures (**i**) and membranous compartments (possibly vesicles) in close proximity to the nuclear envelope (**j**) indicated by black arrowheads. Further examples in Supplementary Fig. 5. Scale bars: 250 nm

control mechanism, acting as a barrier to membrane flow into and out of the nuclear envelope and that the ER protein Lnp1 acts as a secondary barrier, compensating for lack of Lem2.

## Results

**Lem2 deletion augments nuclear size enlargement phenotypes.** The N/C ratio phenotypes of fission yeast cells with mutations in genes encoding inner nuclear membrane proteins were determined using the deletion mutants *ima1Δ, lem2Δ, man1Δ,* and *kms1Δ*[12] and the temperature-sensitive mutant *sad1.1*[13]. None of these mutants displayed a significantly aberrant N/C ratio (Supplementary Fig. 1a). However, deletion of Lem2 augmented the previously characterised nuclear size enlargement of *rae1-167* temperature-sensitive mutant cells (Fig. 1a, b)[11]. *Rae1-167* cells have altered nucleocytoplasmic transport[11,14]. This augmentation was not observed with double mutants of *rae1-167* with mutants of the other inner nuclear membrane proteins (Supplementary Fig. 1a) or other nucleus-localised and organellar membrane-localised proteins tested (Supplementary Fig. 2). Lem2 contains a conserved LEM domain that has been shown to anchor chromatin to the nuclear periphery[15]. We disrupted the chromatin association of Lem2 by deleting its N-terminal helix-extension-helix (HEH) chromatin-binding region[15]. The Lem2 HEH deleted protein failed to augment the *rae1-167* nuclear size enlargement (Fig. 1a), indicating that the role of Lem2 in restricting nuclear enlargement is not dependent on its chromatin binding activity. We also showed that chromatin only occupied part of the enlarged nucleus and thus the extent of chromatin compaction is not affected by the nuclear size changes in *rae1-167 lem2Δ* cells (Fig. 1c). Additionally, we observed that deletion of Lem2 increases the nuclear enlargement observed when nuclear protein export is inhibited by leptomycin B (LMB) (Supplementary Fig. 1b and c). These data indicate that Lem2 functions to restrict the changes in nuclear size that occur following various perturbations, and that these effects are independent of the association of Lem2 with chromatin.

**Lem2 prevents interphase nuclear shrinkage.** Cerulenin is an inhibitor of fatty acid synthetase which thereby reduces cellular membrane availability, and can lead to aberrant mitoses[16]. Treatment of wild type cells with cerulenin results in a slight reduction in the N/C ratio due to these aberrant mitoses but has no effect on the N/C ratio of interphase cells (Fig. 2a–c, Supplementary Fig. 3a, Supplementary Movies 1 and 2). Strikingly, treatment of *lem2Δ* cells with cerulenin leads to rapid shrinkage of interphase nuclei (Fig. 2b, c, Supplementary Movies 3 and 4), resulting in a major reduction in the N/C ratio (Fig. 2a,

Supplementary Fig. 3a). This nuclear shrinkage is accompanied by an increase in concentration of the nuclear envelope marker protein Cut11-GFP within the nuclear envelope (Fig. 2b), and apparent membrane blebbing of the envelope (Fig. 2d, e, Supplementary Fig. 3b, Supplementary Movie 7). Staining cellular lipids with the lipophilic dye DiOC$_6$ demonstrated that changes in cellular lipid distribution and nuclear shape accompany the nuclear shrinkage, with concentrated foci of membrane staining becoming visible on the nuclear envelope (Fig. 2f, Supplementary Fig. 4). We further characterised these membrane changes by transmission electron microscopy. We observed in *lem2Δ* cells that the nuclear membrane remains intact and, as previously reported[15], that the nucleus becomes altered in shape. In *lem2Δ* but not in wild type cells treated with cerulenin, we observed nuclear envelope blebbing events (Fig. 2g, h), the presence of intranuclear membranous structures (Fig. 2i, Supplementary Fig. 5a) and an increased frequency of membrane-bound bodies in close proximity to the nuclear envelope compared with cerulenin-treated wild type cells (mean number per cell slice was 2.7 ($n = 31$ cells) in *lem2Δ* and 1.1 ($n = 27$ cells) in wild type) (Fig. 2j, Supplementary Fig. 5b). These data indicate that the presence of Lem2 prevents nuclear shrinkage during interphase in cerulenin-treated wild type cells, and that in the absence of Lem2, nuclear envelope budding, the formation of intranuclear membranous structures, and interactions between the nuclear envelope and other parts of the cellular membrane system are more frequent.

**Lem2 acts as a barrier to membrane flow.** Given these results we hypothesised that Lem2 acts as a barrier to membrane flow between the nuclear envelope and other parts of the cellular membrane system. This hypothesis was further supported by the observation that the nuclear shrinkage of *lem2Δ* cells treated with cerulenin was suppressed by treatment with brefeldin A (Fig. 3a, b). Brefeldin A inhibits traffic within the golgi and retrograde traffic from the golgi to the ER by blocking COPI recruitment. This results in fusion of golgi and ER membranes and ER expansion[17]. Furthermore, increasing the copy number of Lem2 was found to partially suppress the nuclear size enlargement phenotypes[11] of *rae1-167* and *nem1Δ* cells (Fig. 3c, d), indicating that quantitatively increasing the amount of Lem2 enhances its barrier function. Nem1 is the catalytic subunit of a phosphatase responsible for the dephosphorylation and activation of the lipin family phosphatidic acid phosphatase Ned1[18,19], and its deletion leads to constitutive proliferation of nuclear and ER membranes in interphase. Increasing the copy number of Lem2 supressed the enlarged nuclear size and aberrant nuclear shape phenotypes of

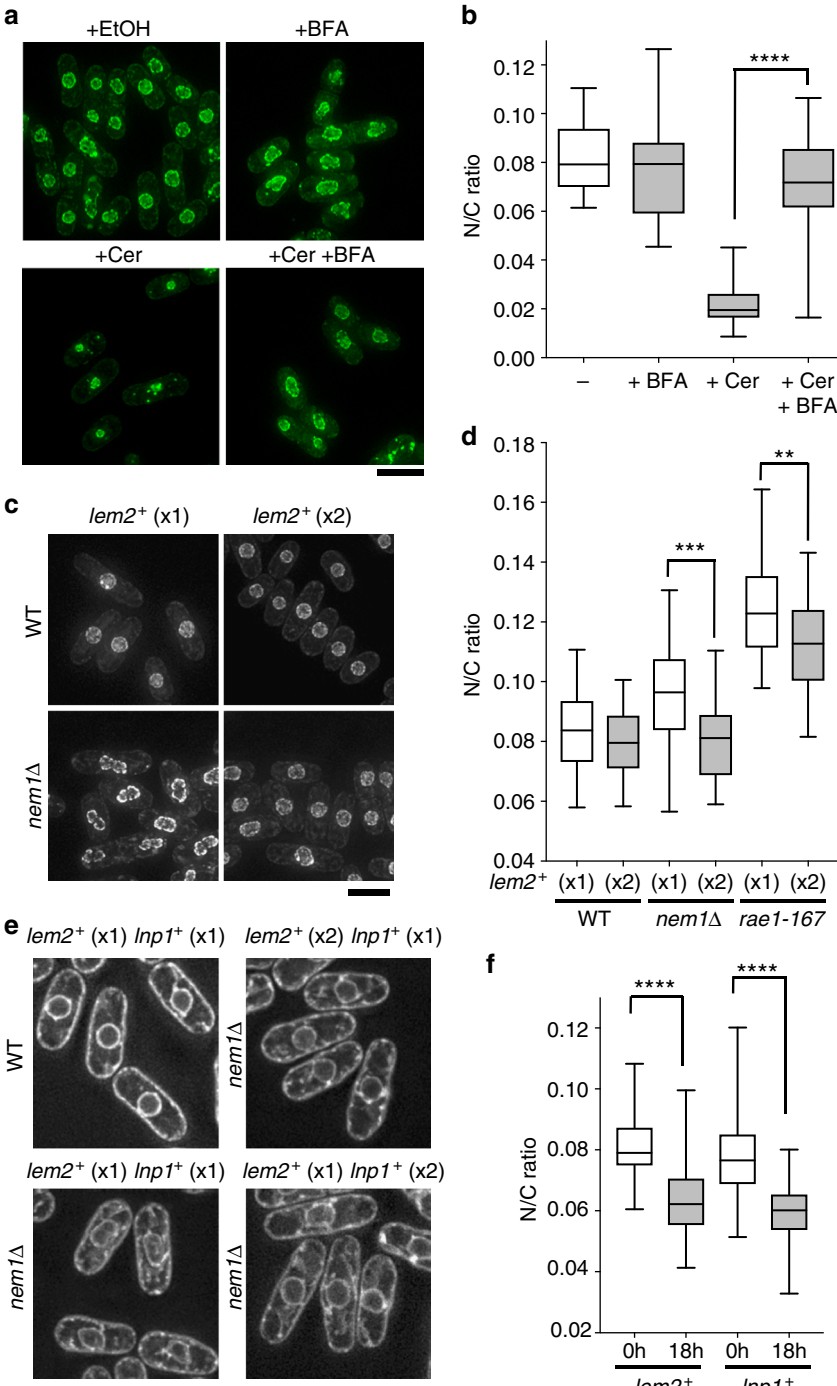

**Fig. 3** Lem2 acts as a barrier preventing inappropriate membrane flow between the nuclear envelope and ER. **a** Images of the nuclear envelope (Cut11-GFP, green) of *lem2Δ* cells grown at 25 °C untreated (+EtOH) or treated with 10 µg/ml cerulenin (+Cer) and/or 100 µg/ml brefeldin A (+BFA) for 3 h. Maximum intensity projections shown. Cerulenin: 10 µg/ml, BFA: 100 µg/ml. Scale bar: 5 µm. **b** N/C ratio of *lem2Δ* cells (25 °C) untreated (−) or treated with 10 µg/ml cerulenin (+Cer) and/or 100 ng/ml brefeldin A (+BFA) for 3 h (n > 30 cells). Two-tailed Mann–Whitney test: ****P < 0.0001, *lem2Δ*+Cer (n = 31 cells), *lem2Δ* +Cer+BFA (n = 32 cells). The corresponding dot plot is available in Supplementary Fig. 9c. **c** Images of the nuclear envelope (Cut11-GFP, grayscale) of wild type (WT) and *nem1Δ* cells containing one or two copies of the gene encoding Lem2 grown at 25 °C. Scale bar: 5 µm. **d** N/C ratio of wild type (WT), *nem1Δ* and *rae1-167* cells containing one or two copies of the gene encoding Lem2 (n > 30 cells). Two-tailed t-tests: ***P = 0.001, *nem1Δ*+1×*lem2+* (n = 31 cells) against *nem1Δ* +2×*lem2+* (n = 40 cells), t = 4.129, degrees of freedom = 69; **P = 0.0015, *rae1-167*+1×*lem2+* (n = 43 cells) against *rae1-167*+2×*lem2+* (n = 35 cells), t = 3.297, degrees of freedom = 76. WT and *nem1Δ* cells grown at 25 °C. Temperature-sensitive *rae1-167* cells grown at 25 °C then shifted to 36 °C for 4 h before imaging. The corresponding dot plot is available in Supplementary Fig. 9d. **e** Representative images of the ER lumen (GFP-ADEL) of wild type (WT) cells and *nem1Δ* cells containing one or two copies of genes encoding Lem2 or Lnp1 grown at 25 °C. Scale bar: 5 µm. **f** N/C ratio of wild type (WT) cells overexpressing *lem2+* or *lnp1+* (under control of the *nmt1* promoter) grown in minimal medium (EMM) at 28 °C for 0 or 18 h (n > 30 cells). Two-tailed Mann–Whitney test: ****P < 0.0001, *lem2+* (0 h) (n = 42 cells) against *lem2+*(18 h) (n = 35 cells); *lnp1+*(0 h) (n = 48 cells) against *lnp1+* (18 h) (n = 48 cells). The corresponding dot plot is available in Supplementary Fig. 9e. In all box-and-whiskers diagrams, boxes indicate median and upper and lower quartile and whiskers indicate range of data

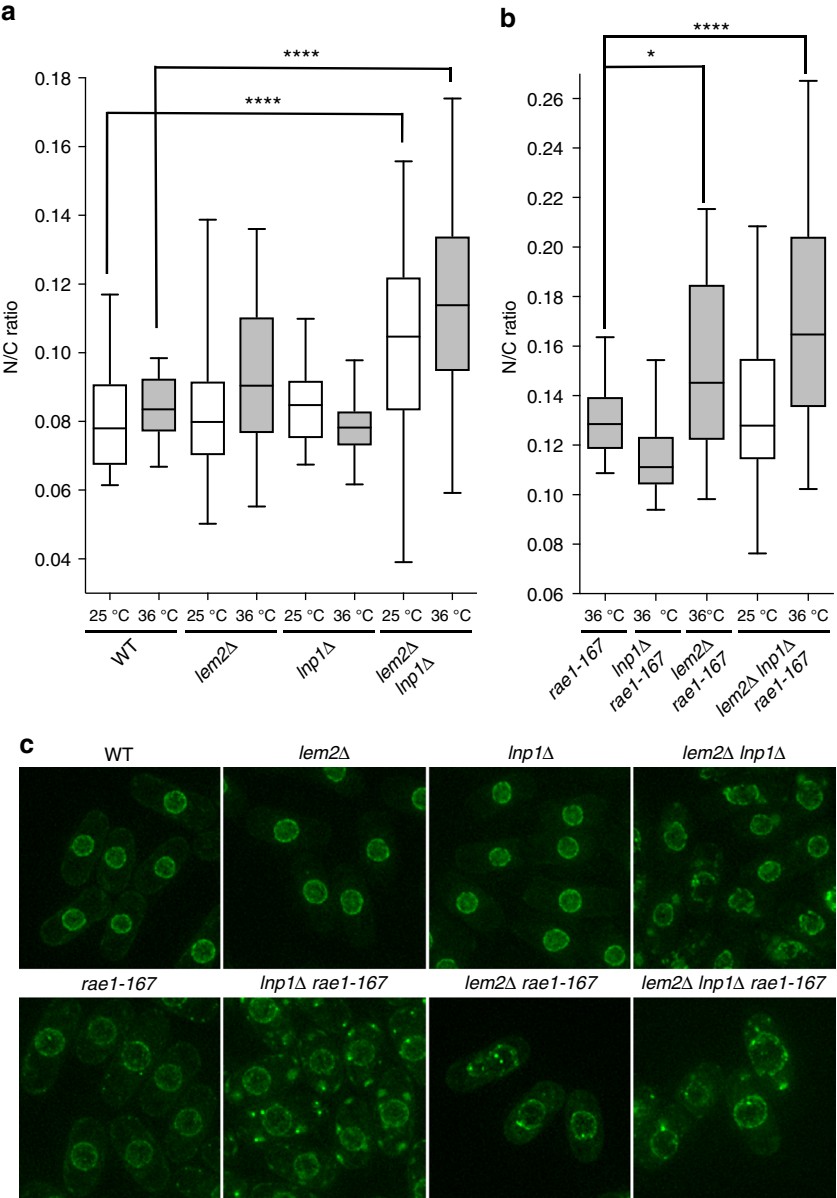

**Fig. 4** Lnp1 compensates for loss of Lem2 to aid preservation of nuclear size. **a** N/C ratio of wild type (WT), *lem2Δ*, *lnp1Δ*, and *lem2Δ lnp1Δ* cells grown at 25 °C then shifted to the indicated temperature for 4 h (*n* > 30 cells). Two-tailed Mann–Whitney test: ****P < 0.0001, *lem2Δ lnp1Δ* (25 °C) (*n* = 32 cells) and *lem2Δ lnp1Δ* (36 °C) (*n* = 54 cells) against WT (*n* = 35 cells); *lnp1Δ* (25 °C) (*n* = 59 cells) and *lnp1Δ* (36 °C) (*n* = 31 cells) against WT (*n* = 35 cells), *P* = 0.0647 and 0.0788, respectively. The corresponding dot plot is available in Supplementary Fig. 9f. **b** N/C ratio of *rae1-167*, *lnp1Δ rae1-167*, *lem2Δ rae1-167*, and *lem2Δ lnp1Δ rae1-167* cells grown at 25 °C then shifted to the indicated temperature for 4 h (*n* > 30 cells). Two-tailed Mann–Whitney test: *P = 0.0139, ****P < 0.0001, *lem2Δ rae1-167* (*n* = 31 cells) and *lem2Δ lnp1Δ rae1-167* (*n* = 31 cells) against *rae1-167* (*n* = 67 cells). The corresponding dot plot is available in Supplementary Fig. 9g. **c** Images of the nuclear envelope (Cut11-GFP, green) of wild type (WT), *lem2Δ*, *lnp1Δ*, *lem2Δ lnp1Δ*, *rae1-167*, *lnp1Δ rae1-167*, *lem2Δ rae1-167*, and *lem2Δ lnp1Δ rae1-167* cells grown at 25 °C then shifted to 36 °C for 4 h. Maximum intensity projections shown. Scale bar: 5 μm. In all box-and-whiskers diagrams, boxes indicate median and upper and lower quartile and whiskers indicate range of data

*nem1Δ* cells (Fig. 3e). Lem2 localisation is not affected by increasing Lem2 copy number although the intensity of Lem2 staining in the nuclear envelope is increased (Supplementary Fig. 6a). Additionally, overexpression of Lem2 in wild type cells was sufficient to lead to a decrease in N/C ratio (Fig. 3f). We observed a further example of nuclear shrinkage in a *lem2Δ* cell after it was born with an enlarged N/C ratio. It underwent rapid nuclear shrinkage, accompanied by what appeared to be an accumulation of vesicles in the cytoplasm, resulting in a rapid reduction of the N/C ratio from 0.138 to 0.116 within 30 min (Supplementary Fig. 6b). Overall these data indicate that Lem2

acts as a barrier to membrane flow between the nuclear envelope and other parts of the cellular membrane system, and that this influences nuclear size.

**ER protein Lnp1 functionally compensates for lack of Lem2.** Lunapark/Lnp1 is an ER protein conserved from yeast to humans that stabilises three-way ER junctions[20,21]. Overexpression of Lnp1 has been shown to suppress the slow growth phenotypes of *lem2Δ*, suggesting functional compensation[22]. We therefore tested whether Lnp1 acts as a secondary barrier to membrane flow. Deletion of *lnp1* did not affect nuclear size when Lem2 was

present, but *lem2Δ lnp1Δ* cells displayed an enlarged N/C ratio (Fig. 4a, c), and deletion of Lnp1 exacerbated the N/C ratio enlargement phenotype of *lem2Δ rae1-167* cells (Fig. 4b, c). Additionally, increasing the copy number of Lnp1 partially suppressed the nuclear size decrease observed in *lem2Δ* cells following inhibition of fatty acid synthesis (Supplementary Fig. 7). As for Lem2, increasing the copy number of Lnp1 suppressed the nuclear size and shape phenotypes of *nem1Δ* cells (Fig. 3e), and overexpression of Lnp1 significantly reduced the N/C ratio of wild type cells (Fig. 3f). These data indicate that Lnp1 is able to partially compensate for Lem2, acting to prevent both over-proliferation and shrinkage of the nuclear envelope by regulating membrane flow within the organellar membrane system. We propose that the regulation of the flow of membrane within the ER affects membrane flow into and out of the nuclear envelope, thus influencing nuclear size.

## Discussion

Our data implicates the inner nuclear membrane protein Lem2 in maintaining an appropriate nuclear size. Lem2 acts as a barrier to membrane flow between organellar compartments, preventing excessive nuclear envelope growth and shrinkage. The ER protein Lnp1 can partially compensate for Lem2, preventing inappropriate membrane flow between the nuclear envelope and other membranous structures. Our study highlights the interconnectedness of cellular membrane compartments, and identifies Lem2 and Lnp1 as barriers to membrane flow, regulating transfer of membrane between compartments, and thus contributing to the control of nuclear size.

This regulation of membrane flow can provide an explanation for the maintenance of nuclear volume as a fixed proportion of cell volume. The different membrane-bound organelles in eukaryotic cells are interconnected to form an overall cellular membrane system[4,5]. We postulate that as cells increase in size, the overall membrane amount within the cellular membrane system increases proportionally. The association of ribosomes with ER membranes could play a role in ensuring this proportionality, given that ribosome number generally scales with overall cell size[23]. Barrier proteins such as Lem2 and Lnp1 could act like valves regulating membrane flow through the cellular membrane network, such that as cellular membrane content increases with cell size, a proportion of the membrane enters the nucleus at a rate determined by the Lem2 barrier protein. If Lem2 is compromised and membrane synthesis is inhibited then membrane flows out of the nucleus (Fig. 2b, c). Thus Lem2 acts as a valve regulating membrane flow into and out of the nucleus contributing to the maintenance of a constant N/C ratio, a control that is influenced by altering membrane components, membrane synthesis, and nucleocytoplasmic transport[1,9–11]. Given the influence of Lnp1 in the ER on nuclear size, similar controls may act throughout the cellular membrane system, maintaining balance in size between different membrane-bound organelles. These controls could also act to maintain a balance between the size of the nucleus and the ER coordinating transcription in the nucleus with translation by the ribosomes resident in the ER.

## Methods

**Yeast general methods**. *S. pombe* strains used are listed in Supplementary Table 1. Standard *S. pombe* media and methods were used[24]. Cells were grown in YE4S medium unless otherwise indicated. Nuclear size phenotypes following LMB and cerulenin treatment were also assessed in EMM medium (Supplementary Fig. 8). Gene deletion mutants from an *S. pombe* near genome-wide haploid gene deletion collection[12] and an *S. pombe* kinase deletion library[25] were used in genetic crosses to generate double mutant strains. Gene tagging was performed by PCR and homologous recombination[26]. The following primers were used for tagged strain constructions: C-terminal GFP tagging of *lem2+* forward, OKK131,

TATGGGAGTGGGTCGGCACAAATACTTTGGATTTCCAAACCGATCGTTC ATTTATTAATACAACTTCCCCTTTACGTGAACGGATCCCCGGGTTAATT AA; reverse, OKK132, ATAAATATATTATTAACTTACTTTTGTATTAGCTA TGAAAAATGTCATAAATACAAAGCTAGAAATACAGTCGAGAATCA GAATTCGAGCTCGTTTAAAC; N-terminal GFP tagging of *lem2+* under *nmt1* promoter forward, OKK220, TTTAATGTAGATAAGGATTCATGTAATTAAT AGCTTATATCAGCTTTATACCATTTTTGATCAGTTTGAATTATTCACCGA ATTCGAGCTCGTTTAAAC; reverse, OKK221, CCCGATTCATGAAGGATCTT TTTCAAGTCTATGACACGAAGATTTCGCAATTCGAAGTTAGGGTCCTCC CAATTGTCCATTTTGTATAGTTCATCCATGC; N-terminal GFP tagging of *lnp1+* under *nmt1* promoter forward, OKK468,
TGCGACTCTCCTAACCATAAGCAAAAATTATTCATTGTCAAGTGAGTC GTTGAAAGGTTAGACGTTATTAATTTATTAACGAATTCGAGCTCGTTTAA AC; reverse, OKK469, TCTAATCACATGTTAGTGACTTTGAGCCTCTAAAAA CTGAGAAATCCTCCTTTACAAACCTTTTGAAAAAACCAGCCCATTTTGTA TAGTTCATCCATGC.

Deletion of the HEH region of Lem2 was carried out using a Quikchange XL kit (Stratagene) and following primers: forward, OKK179, CAATTGGGAGGACCCTAACTTCGAAAGAATACGGAAAAACAAATT; reverse, OKK180, AATTTGTTTTTCCGTATTCTTTCGAAGTTAGGGTCCTCCCAATTG. The *lem2hehΔ* construct or the additional copy of the *lem2+* or *lnp1+* gene, with its own promoter and terminator, was subcloned into the *leu1+* plasmid pJK148[27], and integrated into the genomic *leu1* locus. For drug treatment, LMB (Sigma-Aldrich, 5.5 μg/ml stock in MeOH), cerulenin (Sigma-Aldrich, 1 mg/ml stock in EtOH) and/ or Brefeldin A (Sigma-Aldrich, 10 mg/ml stock in MeOH) was added to the culture at the indicated concentration.

**Microscopy and image analysis**. Fluorescence imaging was carried out using a DeltaVision Elite microscope (Applied Precision) comprised of an Olympus IX71 wide-field inverted fluorescence microscope, an Olympus Plan APO ×60, 1.4 NA oil objective and a Photometrics CoolSNAP HQ2 camera (Roper Scientific) in an IMSOL 'imcubator' Environment Control System. Imaging was carried out at 25 °C unless otherwise indicated. Images were acquired in 0.2, 0.3, or 0.4 μm z-sections over 2.4, 4.4, or 5.4 μm, with a brightfield reference image in the middle of the sample, and deconvolved using SoftWorx (Applied Precision). Images shown are maximum intensity projections of deconvolved images unless otherwise indicated.

For DiOC₆ staining, 5 μg/ml 3,3′-dihexyloxacarbocyanine iodide (DiOC₆) was added to 1 ml of exponential cell culture and incubated at room temperature for 3 min before imaging.

Timelapse microscopy was carried out using a CellASIC ONIX microfluidic platform with Y04C microfluidics plates (Merck). Cells were imaged in the 3.5 μm height chamber in YE4S. Fifty microliters of exponentially growing cells ($1.26 \times 10^6$ cells/ml) was loaded into the plate at a flow rate of 8 psi, untrapped cells were washed out at a flow rate of 5 psi, then a flow rate of 3 psi was maintained for the duration of the experiment.

**Electron microscopy**. Ten micrograms per milliliter of cerulenin was added to exponentially growing *S. pombe* cells at 32 °C. Ninety minutes after drug addition, N/C ratio phenotype was confirmed by light microscopy (Axioscope, Zeiss). Aliquots of *S. pombe* culture were concentrated via pressurised filtration through a 0.45 μm pore membrane filter (Merck Millipore). Cell concentrate was loaded into a flat specimen carrier (Leica Microsystems) and high-pressure frozen using an EM PACT2 high-pressure freezer (Leica Microsystems). For both *lem2Δ* and wild type cell samples, freezing occurred at between 99 and 109 min after drug addition. Vitrified samples were added to a freeze substitution mix (2% osmium tetroxide, 0.1% uranyl acetate, 5% water, in acetone) and held at −80 °C in a freeze substitution unit (EM AFS2, Leica Microsystems) for 14 h. Temperature was increased to −25 °C over 14 h. Samples were incubated in a 1:1 resin (Epon 812, TAAB Laboratories) and acetone mix for 41 h followed by a 2:1 resin and acetone mix for 5 h. Temperature was increased from −25 to 4 °C over 10 h and four changes of 100% Epon 812 resin were performed over the following 72 h. Temperature was increased to 20 °C over 48 min. Samples were removed from the freeze substitution unit and polymerised by incubation at 60 °C for 2 days.

Resin embedded cells were cut into ~70 nm thin sections using a diamond knife (ultra 45°, Diatome) in a Powertome ultramicrotome (RMC Boeckler). Sections were collected onto formvar-coated copper slot grids (EM Resolutions) and stained using 2% aqueous uranyl acetate, followed by Reynold's lead citrate solution. Electron microscopy imaging was performed using a Technai G2 Spirit BioTWIN transmission electron microscope (ThermoFisher Scientific) equipped with an Orius CCD camera (Gatan).

**Nuclear volume, cell volume, and N/C ratio determination**. To determine N/C ratio, >30 interphase cells from log-phase cultures grown in YE4S were manually measured using Image J (NIH)[1]. Cell volume was determined from brightfield reference images and nuclear volume from fluorescence images of the nuclear envelope marker Cut11-GFP. Volumes were calculated based on axial symmetry assuming simple geometries (cell: rod, nucleus: prolate ellipsoid)[1]. At least two independent experiments were performed.

**Statistics**. For data displayed in box-and-whisker diagrams individual data points are shown in dot plots in Supplementary Figs. 9 and 10. For pairwise comparisons, normality of the data of each distribution was determined using the D'Agostino and Pearson test, and then unpaired two-tailed Student's *t*-tests, or Mann–Whitney tests when appropriate, were carried out to determine significance. Details in figure legends.

**Reporting summary**. Further information on experimental design is available in the Nature Research Reporting Summary linked to this article.

### Data availability
The authors declare that the data supporting the findings of this study are available within the paper and its supplementary information files. Further data are available from the corresponding author upon reasonable request.

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

### Acknowledgements
We thank Nurse lab members, especially J. Greenwood, M. Swaffer, and J. Hayles, for helpful discussions and critical comments on the manuscript and Lucy Collinson and Matt Russell for advice on electron microscopy. This work was supported by the Francis Crick Institute [www.crick.ac.uk] (to P.N.), which receives its core funding from Cancer Research UK (FC01121), the UK Medical Research Council (FC01121), and the Wellcome Trust (FC01121). This work was also supported by the Wellcome Trust [grant number 093917] [www.wellcome.ac.uk] (to P.N.), JSPS Postdoctoral Fellowships for Research Abroad (to K.K.), JSPS KAKENHI [grants JP26660089 and 17K07756] [http://www.jsps.go.jp/j-grantsinaid/index.html] (to K.K.), JSPS Program for Advancing Strategic International Networks to Accelerate the Circulation of Talented Researchers (S2902) (to K.K.), the Hiroshima University Education and Research Support Foundation (to K.K.), the Breast Cancer Research Foundation (to P.N.) and The Lord Leonard and Lady Estelle Wolfson Foundation [www.lordandladywolfson.org.uk] (to P.N.). The funders had no role in study design, data collection and analysis, decision to publish, or preparation of the manuscript.

### Author contributions
K.K., H.C. and P.N. conceived and designed the experiments for this study and wrote and revised the manuscript. K.K., H.C., and A.B. carried out the experiments. K.K. and H.C. carried out the data analysis. K.K. and P.N. acquired the funding and P.N. supervised the project.

### Additional information

**Competing interests:** The authors declare no competing interests.

