## [Peer Review File · Nature Communications]

Reviewers' comments:

Reviewer #1 (Remarks to the Author):

This manuscript by Kume et al. describes that nuclear membrane protein Lem2 augmented the nuclear size enlargement of rae1-167 temperature sensitive mutant cells previously reported in Kume et al. (2017).

1. Major criticism is that the model (Figure 4d) is not supported by data presented. Their conclusion that Lem2 and Lnp1 act as a barrier to membrane flow is too speculative.
2. Nucleocytoplasmic transport is altered in the rae1-167 mutant. It is not clearly explained how membrane flow in this mutant is regulated in the presence of Lem2 and how membrane flow is altered in the absence of Lem2 in a way which leads to the augmented nuclear enlargement. It is not obvious how Rae1 is involved in the model (Figure 4d).
3. The authors demonstrate that treatment of lem2 Δ cells with cerulenin, an inhibitor of fatty acid synthetase, leads to a severe reduction of N/C ratio. It is expected that cerulenin reduces amounts of overall lipid in the cell. It is not clear how Lem2 can prevent the nucleus from shrinkage. As one piece of evidence, the authors show that the nuclear shrinkage of cerulenin-treated lem2 Δ cells is suppressed by brefeldin treatment inhibiting ER-Golgi membrane trafficking, but this doesn't provide direct evidence for Lem2 acting as a barrier between the nuclear envelope, ER and other cellular membranes.
4. In this report, lem2 Δ cells were cultured in YE rich medium. It is suggested that lem2 Δ cells cultured in rich medium can induce genome instability, which can introduce unexpected consequences. In addition, phenotypes of lem2 Δ cells depend on nutritional conditions. Thus, it is recommended to confirm that the phenotypes are reproduced in a minimum medium.
5. Some of statistical analyzes are inappropriate. For example, the authors compared multiple populations in Figure 2a. In this case, a statistical method for multiple comparison need to be used, but they used student's t test. They also used student's t test although non-parametric distributions. In these cases, other appropriate statistical tests, such as Mann-Whitney U test or Steel-Dwass test, should be used depending on data. In Figure 2c, they showed the standard error only from 6 samples. For this small number, it is difficult to expect a population, and thus they need to increase the sample size. Although the authors claim statistical significance, distributions of data (e.g., +Cer+BFA in Figure 3b, lem2 Δ lnp1 Δ in Figure 4a and b) are broad with ranges of those data overlap with the control; therefore it is difficult to deduce biological significance. There may be different populations. If so, it may be difficult to conclude causes of those phenotypes as a single factor that the authors are proposing in their model. Taken together, their data interpretation is speculative, and they need more experiments to confirm their model.

Reviewer #2 (Remarks to the Author):

Kume et al. Nature Communications

In this manuscript, the authors address the role of inner nuclear membrane proteins in maintaining the nuclear to cytoplasm ratio in fission yeast. In screening mutants deleted for inner nuclear membrane proteins, the authors uncovered a role for Lem2 in this process. Lem2 anchors chromatin to the nucleus, however, its role in regulating nuclear size is independent of its ability to associate with chromatin. As published studies showed that the overexpression of the ER membrane protein, Lnp1, suppresses the growth defect of lem2 Δ , the authors asked if Lnp1 also

plays a role in these events. Lnp1, localizes to the three-way junctions of the ER and stabilizes newly formed junctions. Interestingly, Lnp1 was found to prevent the overproliferation of the nucleus and its decrease in size that occurs when cells are exposed to perturbation in the absence of Lem2. The authors conclude that Lem2 and Lnp1 modulate nuclear size by regulating the transfer of membrane between compartments. This is a very nice well documented story that addresses an important question. I have a few comments that should be addressed prior to publication.

1) Given that Lnp1 localizes to the junction between the tubular ER and nuclear envelope, it would be interesting to know if Lnp1 co-precipitates with Lem2.

2) The target of BFA is the Arf exchange factor GBF1. Arf activation is required for the recruitment of the COPI coat to the Golgi in yeast and mammalian cells. In mammals, COPI is also recruited to the ERGIC. COPI is required for retrograde traffic from the Golgi to the ER in yeast and retrograde traffic from the ERGIC to the ER in mammalian cells. COPI also mediates traffic from the ERGIC to the Golgi in mammals, as well as Golgi traffic. Therefore, BFA blocks retrograde traffic from the Golgi to the ER (yeast), the ERGIC to the ER (mammals), the late stages of ER-Golgi traffic (mammals), and Golgi traffic. Blocking retrograde Golgi to ER traffic also indirectly blocks ER-Golgi traffic. The authors should explain how they think BFA suppresses nuclear shrinkage in lem2 Δ mutant cells treated with cerulenin. Just saying BFA blocks ER-Golgi traffic is confusing to a reader who does not know how BFA works.

3) Can the authors comment on the puncta in lnp1 Δ mutant cells?

Reviewer #3 (Remarks to the Author):

We think this paper by an acknowledged expert in the field is entirely appropriate for publication, and we have a few suggestions below which could improve the significance of the work.

Major findings of the paper:

- 1) Lem2 deletion increases the nuclear membrane flow.
- 2) Lnp1 compensates for the nuclear membrane flow from ER membrane
- 3) The Lem domain of the Lem2 protein is used to anchor chromatin to the nuclear periphery.
- 4) Deleting HEH domain from Lem2 did not increase the nuclear flow.
- 5) Cerulenin is a fatty acid synthetase inhibitor, when cells are treated with it and Lem2- they found a serious shrinkage and blebbing of the nuclear membrane.
- 6) Lnp1 deletion coupled with Lem2 has shown to be associated with rapid increase in the N/C ratio.
- 7) Lnp1 compensates for the Lem2 role in the nuclear membrane size by preventing the membrane flow.

Concerns/questions

a) from the deletion of Lem2 protein, is there any significant change in the nuclear protein concentration? Maybe some other proteins are accompanying it for membrane control etc.

b) This paper uses a well-established system in Yeast. Deletion of these two proteins (Lem2 and Lnp1) can alter the nuclear size; however is it the same when doing so in a complex organized tissue like muscle cells or brain cells? Maybe some of the signalling events are also contributing to cell and nuclear size control.

c) Proteomics of Lem2- and Lnp1-fraction of the nucleus and ER respectively might reveal some important factors which also contribute to, or are influenced by, deletion of Lem2 and Lnp1.

Reviewers' comments:

Reviewer #1 (Remarks to the Author):

This manuscript by Kume et al. describes that nuclear membrane protein Lem2 augmented the nuclear size enlargement of rae1-167 temperature sensitive mutant cells previously reported in Kume et al. (2017).

1. Major criticism is that the model (Figure 4d) is not supported by data presented. Their conclusion that Lem2 and Lnp1 act as a barrier to membrane flow is too speculative.

It is important to recognise that the regulation of membrane bounded organellar size is an intriguing but poorly investigated subject with only a handful of papers published in the last hundred years, and that there is little understanding of the mechanisms involved. Our proposed model provides a framework for understanding nuclear size regulation, a framework which to date has been largely lacking. It is also novel and should be of interest to the general reader. It is supported by the data we present which can be summarised as follows: (1) Cells lacking the nuclear membrane protein Lem2 display altered nuclear size homeostasis; (2) Strikingly, when membrane synthesis is inhibited, the nucleus in *lem2* Δ cells significantly shrinks, and the concentration of a protein located in the nuclear membrane increases, indicating a rapid reduction in the nuclear membrane in the absence of Lem2; (3) Light microscopy has identified nuclear blebbing and nuclear membrane marked structures moving from the nucleus; (4) Electron microscopic images show membrane changes, including blebbing and the close juxtapositioning of other membrane bounded structures with the nuclear membrane; (5) Overexpression of Lem2 reduces nuclear size; (6) Synergistic interactions between the ER protein Lnp1 and Lem2 link the nucleus with other membranes structures in the cell. All of these data indicate nuclear membrane alterations linked with changes in the nucleus/cell volume (N/C) ratio consistent with our proposed model and we hope that the Reviewer is persuaded that the model is supported by the data we present.

It is possible that the Reviewer meant that it is too speculative to implicate the endoplasmic reticulum as strongly as we did. We did so due to continuity of the endoplasmic reticulum with the nuclear envelope and the localisation of Lnp1 to endoplasmic reticulum 3-way junctions previously reported in the literature. However, as the flow of membrane we propose could also involve other cellular membrane bodies, not just the endoplasmic reticulum, we have altered our discussion in the text to reflect this possible concern of the Reviewer (Lines 20-21, 96-97 and 120-121) and removed Fig 4d (Line 142 and Fig 4d).

As requested by the Editor we have also carried out several new experiments to provide further evidence of the roles of Lem2 and Lnp1 as barriers to membrane flow into and out of the nuclear envelope:

- a) If Lem2 and Lnp1 restrict membrane flow into the nuclear envelope in wild type cells then their overexpression should restrict the nuclear envelope growth that accompanies cell growth and cell cycle progression in wild type cells, giving rise to a reduced N/C ratio phenotype in the cell population. Indeed, the N/C ratio of wild type cells was significantly reduced following overexpression of either Lem2 or Lnp1. We have added this data to the main text (Lines 114-115, 132-133, Fig 3f).
- b) In Fig 3c and d we demonstrated that increasing the copy number of Lem2 is sufficient to suppress the nuclear envelope expansion observed in *nem1* Δ cells. We have gone on to assess the nuclear envelope and endoplasmic reticulum structure (using the lumen marker GFP-ADEL) concomitantly in *nem1* Δ cells with increased copy number of Lem2, and also of Lnp1. Increasing the copy number of either protein

suppresses the increased nuclear size and aberrant nuclear shape phenotypes of *nem1* Δ cells. This supports our hypothesis that Lem2 and Lnp1 restrict membrane flow into the nuclear envelope. We have added this data to the main text (Lines 111-112, 131-132, Fig 3e).

- c) To better characterise membrane flow out of the nuclear envelope during nuclear shrinkage of *lem2* Δ cells treated with cerulenin (described in Fig 2) we have carried out electron microscopy of *lem2* Δ and wild type cells treated with cerulenin. This analysis confirmed previously reported nuclear envelope irregularities in *lem2* Δ cells (Gonzalez et al., 2012), and also allowed us to characterise the membrane changes observed during nuclear shrinkage in more detail. In *lem2* Δ cells, nuclear shrinkage was accompanied by an increased frequency of membrane bounded bodies in close proximity to the nuclear envelope compared to wild type cells, more apparent nuclear envelope budding events, and the presence of intranuclear membranous structures in *lem2* Δ cells. Taken together, these observations support the view that nuclear envelope membrane is altered in *lem2* Δ cells, and are consistent with budding of nuclear envelope into other parts of the cellular membrane system. This does not occur in wild type cells, or does so only at low frequency, supporting our hypothesis that Lem2 prevents membrane flow out of the nucleus in wild type cells. We have added this data to the manuscript (Lines 85-97, Fig. 2g, h, i and j, Supplementary Fig. 5).

2. Nucleocytoplasmic transport is altered in the *rae1-167* mutant. It is not clearly explained how membrane flow in this mutant is regulated in the presence of Lem2 and how membrane flow is altered in the absence of Lem2 in a way which

leads to the augmented nuclear enlargement. It is not obvious how Rae1 is involved in the model (Figure 4d).

Our manuscript was not clear enough in explaining that the *rae1-167* mutant is being used as a tool to perturb nuclear size in these experiments. We do not propose it has a direct role in regulation of membrane flow. We have previously reported our characterisation of the increased N/C ratio of *rae1-167* cells and its dependency on nuclear envelope expansion in detail (Kume et al., 2017). In *rae1-167* cells when Lem2 is present, we propose that Lem2 acts to restrict membrane flow into the nuclear envelope and so restricts nuclear expansion. When this restriction is lifted by Lem2 deletion nuclear expansion is more extreme. We have now explained the use of *rae1-167* more clearly in the manuscript (Lines 53-56).

3. The authors demonstrate that treatment of *lem2Δ* cells with cerulenin, an inhibitor of fatty acid synthetase, leads to a severe reduction of N/C ratio. It is expected that cerulenin reduces amounts of overall lipid in the cell. It is not clear how Lem2 can prevent the nucleus from shrinkage. As one piece of evidence, the authors show that the nuclear shrinkage of cerulenin-treated *lem2Δ* cells is suppressed by brefeldin treatment inhibiting ER-Golgi membrane trafficking, but this doesn't provide direct evidence for Lem2 acting as a barrier between the nuclear envelope, ER and other cellular membranes.

Our experiment describing the suppression of nuclear shrinkage that accompanies cerulenin treatment in *lem2Δ* cells (Fig 3b) is consistent with Lem2 acting as a barrier to membrane flow between the nuclear envelope and other cellular membrane compartments. However, it is only one of the lines of evidence that we have provided in the paper, and in this revised

version we have provided additional evidence, using different approaches and techniques, to support our proposal. These are summarised here: We demonstrate that the nuclear shrinkage observed in *lem2Δ* cells treated with cerulenin is accompanied by nuclear envelope blebbing (Fig 2d, e, g and h, Supplementary Fig 3b, Supplementary Videos S2c). We observe increasing concentration of nuclear envelope protein Cut11 during nuclear shrinkage (Fig 2b) which demonstrates that distance between neighbouring proteins is reduced and so it is likely that lipid is flowing out of the nuclear envelope. We show there is an increased frequency of both intranuclear membranous structures and membrane bound bodies in close proximity to the nuclear envelope in cerulenin treated *lem2Δ* cells compared to cerulenin treated wild type cells (Fig 2i and j, Supplementary Fig 5). We demonstrate that Lem2 acts as a barrier to membrane flow into the nuclear envelope by showing that its overexpression leads to N/C ratio reduction in wild type cells (Fig 3f), increasing its copy number suppresses the N/C ratio increase of *rae1-167* cells (Fig 3d), and the nuclear size and shape defects of *nem1Δ* cells (Fig 3d and e). The deletion of Lem2 augments the N/C ratio increase of *rae1-167* and LMB treated cells (Fig 1a and Supplementary Fig 1c).

We believe that taken together these data provide good evidence for a role of Lem2 as a barrier to membrane flow into and out of the nuclear envelope important for N/C ratio homeostasis. No experiments were proposed by the Reviewer that we should carry out to test the hypothesis that Lem2 acts as a barrier to membrane flow more directly. We have considered the use of FRAP, but as we have evidence that membrane proteins do not move with the lipid we would need a stable fluorescent marker of the lipids themselves to allow us to carry out an experiment of this sort. This approach is not straightforward. Commercially available lipid dyes such as DiOC6 rapidly bleach during imaging making accurate

quantification of their intensity, particularly in a dynamic experiment such as FRAP, extremely challenging.

4. In this report, *lem2Δ* cells were cultured in YE rich medium. It is suggested that *lem2Δ* cells cultured in rich medium can induce genome instability, which can introduce unexpected consequences. In addition, phenotypes of *lem2Δ* cells depend on nutritional conditions. Thus, it is recommended to confirm that the phenotypes are reproduced in a minimum medium.

To address this issue, we have confirmed that cerulenin and LMB treatment both lead to a greater change of N/C ratio in *lem2Δ* cells than in wild type cells in EMM minimal medium, similar to what we observed in YE4S medium (Additional Fig 1). Therefore the phenotype is also observed in minimal medium.

5. Some of statistical analyzes are inappropriate. For example, the authors compared multiple populations in Figure 2a. In this case, a statistical method for multiple comparison need to be used, but they used student's t test. They also used student's t test although non-parametric distributions. In these cases, other appropriate statistical tests, such as Mann-Whitney U test or Steel-Dwass test, should be used depending on data. In Figure 2c, they showed the standard error only from 6 samples. For this small number, it is difficult to expect a population, and thus they need to increase the sample size. Although the authors claim statistical significance, distributions of data (e.g., +Cer+BFA in Figure 3b, *lem2Δlnp1Δ* in Figure 4a and b) are broad with ranges of those data overlap with the control; therefore it is difficult to deduce biological significance.

There may be different populations. If so, it may be difficult to conclude causes of those phenotypes as a single factor that the authors are proposing in their model.

We thank the Reviewer for highlighting these statistical issues and have corrected them. Unfortunately, the original Fig 2a was mislabelled, and as a consequence it could have suggested that multiple populations were compared, which was not the case. Actually, two pairwise comparisons were carried out. We have corrected this labelling (Fig 2a). For all distributions we have carried out normality testing using the D'Agostino and Pearson test. When a non-parametric distribution was identified we have changed the test used to determine significance of difference between populations to a Mann-Whitney test as suggested (Fig 2a, 3b, 4a and 4b, Supplementary Fig 1a and c, Supplementary Fig 6). We have increased the n value in Fig 2c to 15 cells per strain. We did not identify multiple populations of cells in the data in Fig 3b, 4a or 4b, however we appreciate this is difficult to deduce from box and whisker diagrams, so we have included dot plots of for all box and whiskers plots so that the Reviewer can assess the data more easily (Additional Fig 2).

6. Taken together, their data interpretation is speculative, and they need more experiments to confirm their model.

Following the Reviewers' suggestions, we have added new experimental data to our manuscript providing further lines of evidence for the model we propose. The model we propose is novel, and it is a reasonable proposal in an important area of cell biology that has barely been investigated. However, in light of the Reviewers' comments we have reduced our focus on membrane flow specifically into the endoplasmic reticulum as we argued earlier (20-21, 96-97, 119-121 and 142 and Fig 4d).

Reviewer #2 (Remarks to the Author):

Kume et al. Nature Communications

In this manuscript, the authors address the role of inner nuclear membrane proteins in maintaining the nuclear to cytoplasm ratio in fission yeast. In screening mutants deleted for inner nuclear membrane proteins, the authors uncovered a role for Lem2 in this process. Lem2 anchors chromatin to the nucleus, however, its role in regulating nuclear size is independent of its ability to associate with chromatin. As published studies showed that the overexpression of the ER membrane protein, Lnp1, suppresses the growth defect of *lem2*Δ the authors asked if Lnp1 also plays a role in these events. Lnp1, localizes to the three-way junctions of the ER and stabilizes newly formed junctions. Interestingly, Lnp1 was found to prevent the overproliferation of the nucleus and its decrease in size that occurs when cells are exposed to perturbation in the absence of Lem2. The authors conclude that Lem2 and Lnp1 modulate nuclear size by regulating the transfer of membrane between compartments. This is a very nice well documented story that addresses an important question. I have a few comments that should be addressed prior to publication.

1) Given that Lnp1 localizes to the junction between the tubular ER and nuclear envelope, it would be interesting to know if Lnp1 co-precipitates with Lem2.

The possibility of a physical interaction between Lem2 and Lnp1, though not implied by or required for our model, is an interesting one. As suggested by the Reviewer, we have carried out the Co-IP but did not detect an interaction between Lem2 and Lnp1 (Additional Fig 3). However we were able to detect the previously reported (Vo et al., 2016, Hiraoka et al., 2011) interaction between Lem2 and transmembrane protein Man1. A physical interaction between Lem2 and Lnp1 was not identified by the previously reported proteome-wide yeast two-hybrid interactome analysis (Vo et al., 2016). We have not put this data in the manuscript but have provided it as additional information for the Reviewer.

2) The target of BFA is the Arf exchange factor GBF1. Arf activation is required for the recruitment of the COPI coat to the Golgi in yeast and mammalian cells. In mammals, COPI is also recruited to the ERGIC. COPI is required for retrograde traffic from the Golgi to the ER in yeast and retrograde traffic from the ERGIC to the ER in mammalian cells. COPI also mediates traffic from the ERGIC to the Golgi in mammals, as well as Golgi traffic. Therefore, BFA blocks retrograde traffic from the Golgi to the ER (yeast), the ERGIC to the ER (mammals), the late stages of ER-Golgi traffic (mammals), and Golgi traffic. Blocking retrograde Golgi to ER traffic also indirectly blocks ER-Golgi traffic. The authors should explain how they think BFA suppresses nuclear shrinkage in lem2Δ mutant cells treated with cerulenin. Just saying BFA blocks ER-Golgi traffic is confusing to a reader who does not know how BFA works.

We thank the Reviewer for pointing out that our description was unclear, and have expanded our discussion of the effects of BFA treatment and how it may suppress nuclear shrinkage in the main text (Lines 100-105).

3) Can the authors comment on the puncta in *lnp1Δ* mutant cells?

We agree with the Reviewer that Cut11-GFP puncta are observed in the cytoplasm of *rae1-167 lnp1Δ* cells at 36°C. Possible explanations include that they could be aggregates of mislocalised proteins, or given that Cut11 is a transmembrane protein, they could reflect a membranous structure. As the outer nuclear envelope is continuous with the endoplasmic reticulum they might be a structure of the endoplasmic reticulum, perhaps the three-way junctions to which Lnp1 is reported to localise in wild type cells (Chen et al., 2015). To investigate whether the puncta of Cut11 observed overlap with the endoplasmic reticulum we assessed endoplasmic reticulum lumen (ADEL-GFP) and Cut11 (Cut11-mCherry) distribution concomitantly (Additional Fig 4). Though some Cut11 puncta did overlap with ADEL-GFP suggesting endoplasmic reticulum localisation, there were also others that did not, so the Cut11 puncta that we observe in *rae1-167 lnp1Δ* cells are not universally localised to the endoplasmic reticulum. This will require future investigation and we have not included this data in the manuscript but have provided it as additional information for the Reviewer.

Reviewer #3 (Remarks to the Author):

We think this paper by an acknowledged expert in the field is entirely appropriate for publication, and we have a few suggestions below which could improve the significance

of the work.

Major findings of the paper:

- 1) Lem2 deletion increases the nuclear membrane flow.
- 2) Lnp1 compensates for the nuclear membrane flow from ER membrane
- 3) The Lem domain of the Lem2 protein is used to anchor chromatin to the nuclear periphery.
- 4) Deleting HEH domain from Lem2 did not increase the nuclear flow.
- 5) Cerulenin is a fatty acid synthetase inhibitor, when cells are treated with it and Lem2- they found a serious shrinkage and blebbing of the nuclear membrane.
- 6) Lnp1 deletion coupled with Lem2 has shown to be associated with rapid increase in the N/C ratio.
- 7) Lnp1 compensates for the Lem2 role in the nuclear membrane size by preventing the membrane flow.

Concerns/questions

- a) from the deletion of Lem2 protein, is there any significant change in the nuclear protein concentration? Maybe some other proteins are accompanying it for membrane control etc.

To address the Reviewer's concern that nuclear protein concentration may be affected by deletion of Lem2, we assessed cellular protein distribution using the fluorescent protein-

staining dye fluorescein isothiocyanate (FITC). DAPI staining was used to identify nuclei.

We have previously characterised the nuclear protein accumulation that follows shift of *rae1-167* cells to the restrictive temperature (Kume et al., 2017) so this was used a positive control for nuclear protein accumulation. We did not observe nuclear protein accumulation in cells in which Lem2 was deleted (Additional Fig 5). This data has not been included in the manuscript but is provided for the Reviewer.

To address whether or not proteins other than Lem2 may be involved, we constructed double mutants of *rae1-167* with a large number of other proteins reported to localise to the nucleus and/or to the endoplasmic reticulum, and assessed whether any of these deletions, like deletion of Lem2, augmented the N/C ratio increase of *rae1-167* cells. None of these deletion mutants augmented the N/C ratio defect of *rae1-167* cells suggesting Lem2 action is rather specific. We have added this data to the manuscript (Lines 56-59 and Supplementary Fig 2).

b) This paper uses a well-established system in Yeast. Deletion of these two proteins (Lem2 and Lnp1) can alter the nuclear size; however is it the same when doing so in a complex organized tissue like muscle cells or brain cells? Maybe some of the signalling events are also contributing to cell and nuclear size control.

This is an interesting question, especially in light of the reported roles of Lem2 in ERK signalling during myoblast differentiation (Huber et al., 2009). However study of the role of Lem2 and Lnp1 in nuclear size control in mammalian tissues is beyond the scope of the present manuscript.

c) Proteomics of Lem2- and Lnp1-fraction of the nucleus and ER respectively might

reveal some important factors which also contribute to, or are influenced by, deletion of Lem2 and Lnp1.

We agree that proteomics of fractionated nuclei and endoplasmic reticulum would indeed be of interest and could potentially reveal novel roles for other factors contributing to nuclear size control. However, clean fractionation of the nucleus from the endoplasmic reticulum in yeast is not yet technically possible (Wuhr et al., 2015). In previous work we have generated nuclear enriched fractions for proteomic analysis (Kume et al., 2017), however these are likely to contain both the nuclear envelope to which Lem2 localises and also the endoplasmic reticulum to which Lnp1 localises, so would not give us the resolution required to separate the two fractions.

To address the role of other factors cooperating with Lem2 and Lnp1 in nuclear size control we have instead carried out the double mutant analysis described above (Lines 56-59 and Supplementary Fig 2) to screen for deletion mutants that augment the *rae1-167* N/C ratio enlargement. This data is now presented in the revised manuscript. We did not identify any further factors influencing nuclear size control in this analysis.

References

- CHEN, S., DESAI, T., MCNEW, J. A., GERARD, P., NOVICK, P. J. & FERRO-NOVICK, S. 2015. Lunapark stabilizes nascent three-way junctions in the endoplasmic reticulum. *Proc Natl Acad Sci U S A*, 112, 418-23.
- GONZALEZ, Y., SAITO, A. & SAZER, S. 2012. Fission yeast Lem2 and Man1 perform fundamental functions of the animal cell nuclear lamina. *Nucleus*, 3, 60-76.
- HIRAOKA, Y., MAEKAWA, H., ASAKAWA, H., CHIKASHIGE, Y., KOJIDANI, T., OSAKADA, H., MATSUDA, A. & HARAGUCHI, T. 2011. Inner nuclear membrane protein Ima1 is dispensable for intranuclear positioning of centromeres. *Genes Cells*, 16, 1000-11.

- HUBER, M. D., GUAN, T. & GERACE, L. 2009. Overlapping functions of nuclear envelope proteins NET25 (Lem2) and emerin in regulation of extracellular signal-regulated kinase signaling in myoblast differentiation. *Mol Cell Biol*, 29, 5718-28.
- KUME, K., CANTWELL, H., NEUMANN, F. R., JONES, A. W., SNIJDERS, A. P. & NURSE, P. 2017. A systematic genomic screen implicates nucleocytoplasmic transport and membrane growth in nuclear size control. *PLoS Genet*, 13, e1006767.
- VO, T. V., DAS, J., MEYER, M. J., CORDERO, N. A., AKTURK, N., WEI, X., FAIR, B. J., DEGATANO, A. G., FRAGOZA, R., LIU, L. G., MATSUYAMA, A., TRICKEY, M., HORIBATA, S., GRIMSON, A., YAMANO, H., YOSHIDA, M., ROTH, F. P., PLEISS, J. A., XIA, Y. & YU, H. 2016. A Proteome-wide Fission Yeast Interactome Reveals Network Evolution Principles from Yeasts to Human. *Cell*, 164, 310-323.
- WUHR, M., GUTTLER, T., PESHKIN, L., MCALISTER, G. C., SONNETT, M., ISHIHARA, K., GROEN, A. C., PRESLER, M., ERICKSON, B. K., MITCHISON, T. J., KIRSCHNER, M. W. & GYGI, S. P. 2015. The Nuclear Proteome of a Vertebrate. *Curr Biol*, 25, 2663-71.

Additional Figure 1

(a) N/C ratio of wild type (WT), and *lem2Δ* cells grown in EMM medium with or without 100 ng/ml leptomycin B (+LMB) for three hours or four hours (25°C) (n > 22).

(b) N/C ratio of WT and *lem2Δ* cells grown in EMM medium with or without 10 µg/ml cerulenin for three, four or six hours (25°C) (n > 30).

a (Fig. 1a)

b (Fig. 2a)

c (Fig. 3b)

d (Fig. 3d)

e (Fig. 3f)

f (Fig. 4a)

g (Fig. 4b)

h (Fig. S1a)

i (Fig. S1c)

j (Fig. S5)

Additional Figure 3

(a and b) Immunoprecipitation between (a) Man1-GFP and Lem2-FLAG or (b) Lnp1-GFP and Lem2-FLAG in wild type (25°C). Cell extracts were immunoprecipitated with anti-FLAG antibody and the precipitated proteins (IP) were detected by immunoblotting with anti-GFP or anti-FLAG antibody.

Inp1Δ rae1-167

Additional Figure 4

Images of the nuclear envelope (Cut11-mCherry, red) and the ER lumen (GFP-AEDL, green) of *rae1-167 inp1Δ* cells grown at 25°C then shifted to 36°C for four hours. Maximum intensity projection (upper) and single section (bottom) are shown. Scale bar: 5 μm.

Additional Figure 5

Representative images of protein distribution (FITC staining) in *lem2Δ* cells grown at 25°C. DAPI staining was carried out to determine nuclear location. FITC and DAPI staining of *rae1-167* cells at permissive (25°C) and restrictive (36°C, 1 h) temperature are included as a positive control for nuclear protein accumulation. Nuclear accumulation of protein is observed in *rae1-167* cells grown at 36°C for 1h but not in those grown at 25°C (Kume *et al.* 2017) . Scale bar: 10 μm.

REVIEWERS' COMMENTS:

Reviewer #1 (Remarks to the Author):

The manuscript has been greatly improved through the revision with additional experiments. Now I have only minor suggestions for publication.

1. The authors provided statistical analyses as a dot plot only for reviewers. These data are helpful for readers to see statistical distribution of the data if presented in the main figures or additionally in supplementary figures.
2. As for Figure 3f, the authors should state their experimental conditions more clearly. It seems that the nmt1 promoter was used for overexpression of Lem2 and Lnp1 judging from the strain list, but it is not stated anywhere. In this relation, it is difficult to find the strains used in each experiment. Thus, in the strain list, it is recommended to specify which figures the strains were used for.
3. I would recommend to present "Additional Figure1" as a supplementary figure, which will be helpful for readers.
4. In Figure 2c, label the axes and the unit (for the vertical axis, the cell volume in μm^3 ? - also for the horizontal axis).
5. It seems that most of the deletion mutants listed in Supplementary Table 1 were made by crossing the rae1-167 strain (FN280) with a previously made library of deletion strains. If so, this should be mentioned somewhere in the text (e.g., methods).

Response to reviewers
Responses italicised.

Reviewer #1 (Remarks to the Author):

The manuscript has been greatly improved through the revision with additional experiments. Now I have only minor suggestions for publication.

1. The authors provided statistical analyses as a dot plot only for reviewers. These data are helpful for readers to see statistical distribution of the data if presented in the main figures or additionally in supplementary figures.

These data have been added to the manuscript as Supplementary Figure 9 and Supplementary Figure 10 and are referred to in the Methods section (lines 320-321).

2. As for Figure 3f, the authors should state their experimental conditions more clearly. It seems that the nmt1 promoter was used for overexpression of Lem2 and Lnp1 judging from the strain list, but it is not stated anywhere. In this relation, it is difficult to find the strains used in each experiment. Thus, in the strain list, it is recommended to specify which figures the strains were used for.

As suggested, we have added this information to the legend of Figure 3 (lines 493-494). We have also added a column specifying which figures the strains were used for to Supplementary Table 1.

3. I would recommend to present “Additional Figure1” as a supplementary figure, which will be helpful for readers.

We have added this Figure to the manuscript as Supplementary Figure 8 and refer to it in the Methods section (lines 228-230).

4. In Figure 2c, label the axes and the unit (for the vertical axis, the cell volume in μm^3 ? - also for the horizontal axis).

We apologise for this oversight; the axis labels were lost in conversion to PDF. We have corrected this.

5. It seems that most of the deletion mutants listed in Supplementary Table 1 were made by crossing the rae1-167 strain (FN280) with a previously made library of deletion strains. If so, this should be mentioned somewhere in the text (e.g., methods).

As suggested, we have added this information to the Methods section (lines 230-232).